

# Global ecosystem-scale plant hydraulic traits retrieved using model-data fusion

Yanlan Liu[1], Nataniel M. Holtzman[1], and Alexandra G. Konings[1]

[1]Department of Earth System Science, Stanford University, Stanford, CA, USA 94305

**Correspondence:** Yanlan Liu (liu.9367@osu.edu)

**Abstract.** Droughts are expected to become more frequent and severe under climate change, increasing the need for accurate predictions of plant drought response. This response varies substantially depending on plant properties that regulate water transport and storage within plants, i.e., plant hydraulic traits. It is therefore crucial to map plant hydraulic traits at a large scale to better assess drought impacts. Improved understanding of global variations in plant hydraulic traits is also needed for parameterizing the latest generation of land surface models, many of which explicitly simulate plant hydraulic processes for the first time. Here, we use a model-data fusion approach to evaluate the spatial pattern of plant hydraulic traits across the globe. This approach integrates a plant hydraulic model with datasets derived from microwave remote sensing that inform ecosystem-scale plant water regulation. In particular, we use both surface soil moisture and vegetation optical depth (VOD) derived from the X-band JAXA Advanced Microwave Scanning Radiometer for EOS (AMSR-E). VOD is proportional to vegetation water content and therefore closely related to leaf water potential. In addition, evapotranspiration (ET) from the Atmosphere Land-Exchange Inverse model (ALEXI) is also used as a constraint to derive plant hydraulic traits. The derived traits are compared to independent data sources based on ground measurements. Using the K-means clustering method, we build six hydraulic functional types (HFTs) with distinct trait combinations - mathematically tractable alternatives to the common approach of assigning plant hydraulic values based on plant functional types. Using traits averaged by HFTs rather than by PFTs improves VOD and ET estimation accuracies in the majority of areas across the globe. The use of HFTs and/or plant hydraulic traits derived from model-data fusion in this study will contribute to improved parameterization of plant hydraulics in large-scale models and the prediction of ecosystem drought response.

## 1 Introduction

Water stress during drought restricts photosynthesis, thus weakening the strength of the terrestrial carbon sink (Ma et al., 2012; Wolf et al., 2016; Konings et al., 2017) and possibly causing plant mortality under severe conditions (McDowell et al., 2016; Adams et al., 2017; Choat et al., 2018). The plant response to water stress also directly controls regional water resources and drought propagation by modulating water flux and energy partitioning between the land surface and the atmosphere (Goulden and Bales, 2014; Manoli et al., 2016; Anderegg et al., 2019). However, how plants regulate water, carbon and energy fluxes and plant mortality under drought could vary considerably depending on plant properties, particularly plant hydraulic traits





(Sack et al., 2016; Hartmann et al., 2018; McDowell et al., 2019). Understanding this variation is therefore crucial to accurate prediction of ecosystem dynamics under changing climate.

Plant hydraulic traits at both stem (e.g., $\psi_{50,x}$, the xylem water potential under 50% loss of xylem conductivity) and stomatal (e.g, $g_1$, the sensitivity parameter of stomatal conductance to vapor pressure deficit) levels control plant water uptake and the extent of stomatal closure under water stress (Martin-StPaul et al., 2017; Feng et al., 2017; Meinzer et al., 2017; Anderegg

et al., 2017). Distinct hydraulic traits across species and plant communities define hydraulic strategies, which lead to different responses of leaf water potential and gas exchange during drought (Matheny et al., 2017; Barros et al., 2019). Plant hydraulic traits play critical roles in predicting stomatal response to stress (Sperry et al., 2017; Liu et al., 2020), plant water storage (Huang et al., 2017), leaf desiccation (Blackman et al., 2019), and drought-driven tree mortality risk (Anderegg et al., 2016; Powell et al., 2017; Liu et al., 2017; De Kauwe et al., 2020). As a result of their effect on the surface energy balance, plant

hydraulic traits also impact the magnitude of land-atmosphere feedbacks (Anderegg et al., 2019). In dry tropical forests, leaf water potential - which is directly influenced by hydraulic traits - has also been shown to affect leaf phenology (Xu et al., 2016). As a result, it has been increasingly recognized that plant hydraulic traits are important in mediating ecosystem drought response and hydroclimatic feedbacks at regional to global scales (Choat et al., 2012; Anderegg, 2015; Choat et al., 2018; Hartmann et al., 2018).

Understanding how plant hydraulic traits modulate large-scale drought responses requires mapping these traits. At large scales, plant traits are often parameterized based on plant functional types (PFTs). However, plant hydraulic traits can vary as much across PFTs as within them (Anderegg, 2015; Konings and Gentine, 2017). Finding alternative ways to scale up *in situ* measurements using a bottom up approach is challenging because the spatial coverage of such measurements is often limited and biased towards temperate regions. Furthermore, plant hydraulic traits are highly variable within species (Anderegg, 2015)

and even between different components of a single plant and across vertical gradients within individual trees (Johnson et al., 2016). Alternatively, because microwave remote sensing observations of vegetation optical depth (VOD) are sensitive to leaf water potential (Momen et al., 2017; Konings et al., 2019; Holtzman et al., 2020), they may carry implicit information that can be used to disentangle plant hydraulic traits, without the need for explicit upscaling.

Konings and Gentine (2017) first derived plant hydraulic trait variations at large scales by using VOD to calculate the

effective ecosystem-scale isohydricity. The isohydricity reflects the response of leaf water potential as soil water potential dries down (Tardieu and Simonneau, 1998). At a stand scale, this plant physiological metric has been used to explain photosynthesis variations (Roman et al., 2015) and drought mortality risk (McDowell et al., 2008) across species. At a global scale, remote-sensing derived isohydricity patterns have been used to explain photosynthesis sensitivity to VPD and soil moisture in North American grasslands (Konings et al., 2017) and the Amazon (Giardina et al., 2018), to explore the interannual variability

of isohydricity (Wu et al., 2020), and to explain the relationship between drought resistance and resilience in gymnosperms (Li et al., 2020). However, because isohydricity is an emergent rather than intrinsic property, it is subject to change with environmental conditions (Hochberg et al., 2018; Novick et al., 2019; Feng et al., 2019; Mrad et al., 2019). Furthermore, isohydricity is influenced by both stomatal and xylem traits (Martínez-Vilalta et al., 2014), which do not always co-vary



(Manzoni et al., 2013; Martínez-Vilalta et al., 2014; Bartlett et al., 2016; Martínez-Vilalta and Garcia-Forner, 2017). Estimating
intrinsic xylem and stomatal traits separately is therefore necessary for better assessment of plant drought response.

From a modeling perspective, as plant hydraulics has been increasingly recognized as a central link connecting hydro-
climatic processes and ecosystem ecology (Sack et al., 2016; McDowell et al., 2019), land surface and dynamic vegetation
models that explicitly incorporate plant hydraulics are becoming more common (e.g. Xu et al., 2016; Christoffersen et al.,
2016; Kennedy et al., 2019; De Kauwe et al., 2020; Eller et al., 2020). However, explicit plant hydraulic representation also
requires parametrization choices for the associated plant hydraulic traits. As discussed above, a bottom-up scaling of *in situ*
measurements is likely to miss significant fractions of the spatial variability in these parameters. Alternatively, Liu et al. (2020)
took a top-down inversion approach by integrating a plant hydraulic model with ET data observed at FLUXNET sites. This
model-data fusion approach identifies the most likely traits generating modeled dynamics consistent with observations, thus
providing effective hydraulic traits that represent ecosystem-scale behaviours. Similar model-data fusion approaches have been
previously applied in carbon cycle models (e.g. Wang et al., 2009; Dietze et al., 2013; Quetin et al., 2020). Not surprisingly,
many of these applications suggest that integrating informative observations is among the keys to effectively constraining
model parameters.

Here, we use the model-data fusion approach to evaluate the global pattern of ecosystem-scale plant hydraulic traits. Specif-
ically, we determined global maps of five plant hydraulic traits (see methods). To effectively constrain the traits, we use several
datasets derived from microwave remote sensing observations, each of which is affected by plant hydraulic behaviour. Specif-
ically, we used VOD, surface soil moisture, and ET estimates from a microwave implementation of the Atmosphere Land
Exchange Inverse (ALEXI) framework. The resulting retrieved ecosystem-scale plant hydraulic traits are then compared to
available *in situ* observations. Having derived spatial maps of variations in plant hydraulic traits, we explore whether sim-
ple alternatives to PFTs can be built to facilitate parameterizing land surface models. We derive several so-called 'hydraulic
functional types' (HFTs) based on the clustering of retrieved hydraulic traits and examine their spatial patterns.

## 2 Methods

### 2.1 Plant hydraulics model

For the model underlying the model-data fusion system, we used a soil-plant system model adapted from Liu et al. (2020)
that incorporates plant hydraulics. The soil is characterized by two layers: a hydraulically active rooting zone extending to the
maximum rooting depth, topped by a surface layer with a fixed depth of 5 cm. Soil moisture in both layers is modeled based
on the soil water balance, i.e.,

$$Z_1 \frac{\mathrm{d}s_1}{\mathrm{d}t} = P - L_{12} - E \tag{1}$$

$$Z_2 \frac{\mathrm{d}s_2}{\mathrm{d}t} = L_{12} - L_{23} - J \tag{2}$$

where $Z_1$ ( = 5 cm) and $Z_2$ are the thickness of the two soil layers and $s_1$ and $s_2$ are the volumetric soil moisture of the two
layers. $P$ is the precipitation rate, $E$ is the soil evaporation rate, and $J$ is plant water uptake. The $L_{12}$ and $L_{23}$ are vertical fluxes





between the two soil layers and out of the rooting zone respectively. Both are calculated based on Darcy's law. A constant soil moisture below the rooting zone is assumed as the boundary condition for the $L_{23}$ calculation. The soil evaporation rate $E$ is calculated as the potential evaporation from the Penman equation multiplied by a stress factor of $s_1/n$, where $n$ is the soil porosity. The potential evaporation is driven by the fraction of total net radiation that penetrates through the canopy to the ground surface based on Beer's law (Campbell and Norman, 1998). The remaining fraction of total net radiation is absorbed by the leaves and drives transpiration (Eq.7). Plant water uptake $J$ is determined as the product of the whole-plant conductance ($g_p$) and the water potential gradient between the soil ($\psi_s$) and the leaf ($\psi_l$), i.e.,

$$J = g_p \left( \psi_s - \psi_l \right),$$

(3)

where the soil water potential is calculated from $s_2$ based on the empirical soil water retention curve by Clapp and Hornberger (1978).

$$\psi_s = \psi_{s,sat} \left( s_2/n \right)^{-b_0}.$$

(4)

Above, $\psi_{s,sat}$ is the saturated soil water potential, $n$ is the soil porosity, and $b_0$ is the shape parameter. Plant water uptake from the thin surface layer is assumed to be negligible. The whole-plant conductance varies with leaf water potential following a linear vulnerability curve as

$$g_p = g_{p,max} \left( 1 - \frac{\psi_l}{2\,\psi_{50,x}} \right),$$

(5)

where $g_{p,max}$ is the maximum xylem conductance and $\psi_{50,x}$ is the water potential at which xylem conductance drops to half of its maximum. A linear vulnerability curve is used because the nonlinearity of the vulnerability curve can hardly be identified using the model data-fusion approach even at a much finer scale of flux tower foot print (Liu et al., 2020). The linearized form here keeps the number of parameters minimal.

The model assumes a single water storage pool in the canopy. The size of this pool is recharged by plant water uptake ($J$) and reduced by transpiration ($T$), with a vegetation capacitance parameter $C$ determining the proportionality between that water flux and the corresponding change in plant water potential.

$$C\,\frac{\mathrm{d}\psi_l}{\mathrm{d}t} = J - T$$

(6)

Transpiration is computed using the Penman-Monteith equation.

$$T = \frac{\Delta\,R_{nl} + \rho_a\,c_p\,g_a\,D}{\lambda\,[\Delta + \gamma\,(1 + g_a/g_s)]}$$

(7)

where $\Delta$ is the rate of change of saturated vapor pressure with air temperature; $R_{nl}$ is the fraction of net radiation absorbed by the leaves; $\rho_a$ is the air density; $c_p$ is the specific heat capacity of air; $g_a$ is the aerodynamic conductance; $D$ is the vapor pressure deficit; $\lambda$ is the latent heat of vaporization; $\gamma$ is the psychrometric constant; and $g_s$ is the stomatal conductance to water vapor per unit ground area. The stomatal conductance is calculated using the Medlyn stomatal conductance model (Medlyn et al., 2011), while omitting cuticular and epidermal losses by assuming zero minimum stomatal conductance.

$$g_s = a_0\,\mathrm{LAI}\left( 1 + \frac{g_1}{\sqrt{D}} \right) \frac{A}{c_a}$$

(8)





where $a_0 = 1.6$ is the relative diffusivity of water vapor with respect to $CO_2$; LAI is the leaf area index; $g_1$ is the slope parameter, inversely proportional to the square root of marginal water use-efficiency (Medlyn et al., 2011; Lin et al., 2015); $A$ is the biochemical demand for $CO_2$ calculated using the photosynthesis model (Farquhar et al., 1980); and $c_a$ is the atmospheric $CO_2$

concentration. Photosynthesis is limited by either RuBP regeneration or by the carboxylation rate. Water stress is assumed to restrict photosynthesis under the carboxylation-limited regime through a down-regulated maximum carboxylation rate ($V_{\mathrm{cmax}}$) following Kennedy et al. (2019) and Fisher et al. (2019).

$$V_{\mathrm{cmax}} = \left(1 - \frac{\psi_l}{2\,\psi_{50,s}}\right) V_{\mathrm{cmax,\,w}} \tag{9}$$

where $\psi_{50,s}$ is the leaf water potential when $V_{\mathrm{cmax}}$ drops to half of its maximum value under well-watered conditions ($V_{\mathrm{cmax,\,w}}$).

The model was driven by climate conditions at a 3-hourly scale. To temporally integrate the model, a forward Euler's method was used for computational efficiency, except for the calculation of plant water uptake, for which Eqns. 2 through 6 were linearized at each time step and then solved analytically to ensure numerical stability. The modeled time-series of ET ($E+T$), surface soil moisture ($s_1$) and VOD were compared with microwave remote sensing observations as described below.

## 2.2   Microwave remote sensing constraints

To derive plant hydraulic traits, the model in Section 2.1 was constrained by microwave remote sensing products of VOD and surface soil moisture, as well as by remote-sensing derived ET, all with a spatial resolution of $0.25°$.

### 2.2.1   VOD

We used VOD and surface soil moisture derived from the JAXA Advanced Microwave Scanning Radiometer for EOS (AMSR-E) retrieved by the Land Parameter Retrieval Model (LPRM) (Owe et al., 2008; Vrije Universiteit Amsterdam and NASA

GSFC, 2016). This dataset is based on observations at X-band frequency (10.7 GHz), which is primarily sensitive to water content of the upper canopy layers. Data for 2003-2011 were used. Outliers that are more than three scaled median absolute deviations away from the median were filtered out and attributed to high-frequency noise in the retrievals common to VOD datasets (Konings et al., 2015, 2016). A five-day moving average method was applied to midday and midnight VOD respectively to further diminish noise in the raw data. Both ascending (1:30 AM) and descending (1:30 PM) observations were used,

to enable them to constrain sub-daily variations in plant hydraulic dynamics.

To relate VOD and leaf water potential, we noted that VOD is proportional to vegetation water content (VWC). In turn, VWC is determined by the product of above ground biomass (AGB) and plant relative water content (RWC).

$$VOD = \beta VWC = \beta\,AGB \times RWC \tag{10}$$

where $\beta$ is the scaling parameter depending on the structure and dielectric properties of plants (Kirdiashev et al., 1979).

As in Momen et al. (2017), AGB is represented using linearized relationships of LAI and $\psi_l$ respectively. The relationship between RWC and $\psi_l$ usually follows a Weibull pressure-volume curve. However, it has been successfully linearized in previous theoretical and observational applications (Manzoni et al., 2014; Momen et al., 2017; Konings and Gentine, 2017). Thus, VOD





is modeled as:

$$\text{VOD} = (a + b\,\text{LAI})\,(1 + c\,\psi_l) \tag{11}$$

where $a$ and $b$ are the scaling parameters from LAI to $\beta$AGB; and $c$ is the linearized slope of the pressure-volume curve. The $a$, $b$ and $c$ parameters vary across pixels and were retrieved as additional inversion parameters as part of the model-data fusion process.

### 2.2.2 Soil moisture

We also used the associated surface soil moisture retrievals from LPRM as additional constraints. Instead of performing a direct
comparison between modeled and retrieved soil moisture, we followed the widely-used approach of assimilating retrieved soil moisture only after matching its cumulative distribution function (cdf) to the modeled soil moisture (Reichle and Koster, 2004; Su et al., 2013; Parrens et al., 2014). Because the magnitudes of both retrieved and modeled soil moisture are highly dependent on the retrieval algorithm and specific model structure (Koster et al., 2009), this cdf-matching approach reduces the effect of bias in either the model or observations on the ability of the soil moisture observations to act as useful constraints. Unlike
VOD, surface soil moisture does not have a strong diurnal cycle. Additionally, because the canopy and soil often reach thermal equilibrium at night, AMSR-E retrievals at 1:30 PM have greater retrieval errors than at 1:30 AM (Parinussa et al., 2016). Therefore, only 1:30 AM surface soil moisture was included as a model constraint here.

### 2.2.3 Evapotranspiration

The model was also constrained by weekly ET during 2003-2011. ET was estimated using the Atmosphere–Land Exchange
Inverse (ALEXI) algorithm (Anderson et al., 1997, 2007; Holmes et al., 2018). Most remote sensing-based ET datasets assume prior values of stomatal parameters (Kalma et al., 2008; Wang and Dickinson, 2012), which would make it circular to retrieve plant traits based on these datasets. By contrast, the ALEXI framework is relatively independent from prior assumptions on vegetation properties. To achieve this independence, ALEXI uses a two-source energy balance method and is constrained to be consistent with the boundary layer evolution (Anderson et al., 2007; Holmes et al., 2018). We further used a version of ALEXI
based on microwave-derived land surface temperatures rather than optical ones as in the classic ALEXI implementations. When compared to *in situ* observations, microwave-ALEXI and optical-ALEXI performed similarly (Holmes et al., 2018), but the microwave-based version has the advantage of having more observations because, unlikely optically-derived estimates, it is not limited by cloud cover. The 0.25 degree resolution of the microwave ALEXI product is also more consistent with the other components of our model-data fusion system.

### 180 2.3 Model-data fusion

Plant hydraulic traits and several other model parameters controlling plant hydraulic behaviour were retrieved using a Markov Chain Monte Carlo (MCMC) method, which determined the parameter values that yield model output most consistent with observed constraints. Thirteen parameters were retrieved in total, including five plant hydraulic traits ($g_1$, $\psi_{50,s}$, $C$, $g_{p,max}$,



and $\psi_{50,x}$), three scaling parameters relating VOD to $\psi_l$ ($a$, $b$ and $c$ in Eq. 11), two soil properties (including $b_0$ in Eq. 4

and the subsurface boundary condition of soil moisture in the deepest layer), and three uncertainty values, describing the standard deviation of the observational noise of VOD ($\sigma_{\text{VOD}}$), surface soil moisture ($\sigma_{\text{SM}}$) and ET ($\sigma_{\text{ET}}$), respectively. An adaptive Metropolized independence sampler was used to generate posterior samples (Ji and Schmidler, 2013). This sampling method was designed to facilitate convergence especially for nonlinear models and has been shown effective for retrieving plant hydraulic traits at flux tower sites (Liu et al., 2020). To reduce the dimensionality of the parameter space and facilitate

convergence, the MCMC jointly sampled all parameters except the three scaling parameters of VOD. For these parameters, the optimal values were determined conditional on the rest of the parameters after each sampling step based on least squared error. That is, after each sampling step, the three values were optimized so as to minimize the least-squares difference between observed VOD and the predicted VOD conditional on simulated $\psi_l$ and the optimized parameter values for $a$, $b$, and $c$.

The MCMC also incorporated prior information about parameter ranges and constraints on their realistic combinations. For

$\psi_{50,x}$, a generalized extreme value distribution was used as the prior for the corresponding PFT. The distribution was fitted using measurements of species belonging to each PFT in the TRY database (Kattge et al., 2011). The corresponding PFT of each species was determined based on the PLANTS database (USDA, NRCS, 2020) and the Encyclopedia of Life (Parr et al., 2014). For PFTs not included the TRY database, a distribution fitted using measurements for all species was used as the prior (Fig. S1). We also incorporated a physiological constraint from meta-analysis suggesting stomatal conductance is

down-regulated before substantial xylem embolism occurs (Martin-StPaul et al., 2017; Anderegg et al., 2017), i.e.,

$$|\psi_{50,s}| < |\psi_{50,x}|. \tag{12}$$

The physiological constraint, which was also used in Liu et al. (2020), avoids unrealistic combinations of parameters that nevertheless match data. For other parameters, uniform non-informative priors spanning realistic ranges were used (Table S1).

The cost function in the MCMC (i.e., the reverse of the likelihood function multiplied by the prior) determines the esti-

mated posterior distribution of parameters. The likelihood function was calculated by comparing the modeled VOD, surface soil moisture and ET with the three categories of observations. Observations on rainy (daily cumulative precipitation > 1 cm) or freezing (daily minimum air temperature < 0 °C) days were removed. Each of the remaining observations was considered independent, following a Gaussian distribution with a mean of the modeled value and the standard deviation of the corresponding category (i.e., one of $\sigma_{\text{VOD}}$, $\sigma_{\text{SM}}$ and $\sigma_{\text{ET}}$). The likelihood of all observations were then combined after re-weighting each

constraint based on its number of observations. That is,

$$\log\left(L(y_v^{(1:n_v)}, y_e^{(1:n_e)}, y_s^{(1:n_s)} | \theta)\right) = \left(\frac{1}{n_v}\sum_{i=0}^{n_v}\log L(y_v^{(i)}|\theta) + \frac{1}{n_e}\sum_{i=0}^{n_e}\log L(y_e^{(i)}|\theta) + \frac{1}{n_s}\sum_{i=0}^{n_s}\log L(y_s^{(i)}|\theta)\right)\frac{n_v+n_e+n_s}{3} \tag{13}$$

where $L$ is likelihood of observed VOD ($y_v$), ET ($y_e$), and surface soil moisture ($y_s$) under given parameters $\theta$ (including all the thirteen parameters to be retrieved); $n_v$, $n_e$, and $n_s$ are the number of valid data of VOD, ET and surface soil moisture, respectively. Due to the unbalanced number of observations among the measurement types, re-normalizing the weights in each

category based on its number of observations avoids over-weighting of semi-daily VOD and surface soil moisture over weekly ET observations.



For the global retrievals, pixels classified by MODIS land cover data as wetland, urban area, barren area, snow/ice covered, or tundra-dominated were excluded from the analysis. Pixels where VOD is below 0.15 or above 0.8 were also excluded to remove sparsely vegetated pixels and extremely dense vegetation areas, respectively. The most densely vegetated areas were
removed because low microwave transmissivity significantly reduces the accuracy of VOD and soil moisture retrievals there (Kumar et al., 2020), and low VOD pixels were removed to reduce inaccuracies due to ground volume scattering and low vegetation density. For the remaining pixels, parameters were retrieved using observations in 2004 and 2005, during which the El Niño event and the elevated tropical North Atlantic sea surface temperatures induced drought stress in many regions across the globe (Phillips et al., 2009; FAO, 2014). Here, we used only two years of observations, rather than the entire period, to
reduce the computational load of model-data fusion. The remaining seven years were used for testing. Separating retrieval and testing periods also helped to (potentially) identify overfitting.

For each pixel, four MCMC chains were used. Each started randomly within the prior parameter ranges, and each generated 50,000 samples. Within- and among-chain convergences were diagnosed by Gelman–Rubin (<1.2) and Geweke values (<0.2) (Brooks and Gelman, 1998). Across the studied pixels, all parameters converged for 79% of pixels, while at least half of
the parameters converged for 97% of pixels. The remaining 3% of pixels that did not converge were removed from analysis. For each pixel, 200 samples were randomly selected from the chains after step 40,000 as posterior samples of parameters. Ensemble means of VOD, surface soil moisture, and ET modeled using posterior samples were compared to observations during the period 2003-2011. Posterior means of the hydraulic traits in each pixel were used for analysis below.

## 2.4 Climate forcing and ancillary properties

The model-data fusion system was run at 0.25° resolution. Meteorological drivers at this spatial resolution and the 3-hourly temporal resolution used by the model were derived from the Global Land Data Assimilation System (GLDAS) (Rodell et al., 2004; Beaudoing and Rodell, 2020). In particular, GLDAS-derived forcings include net shortwave radiation, air temperature, precipitation, surface atmospheric pressure, specific humidity, and aerodynamic conductance calculated using the ratio between the sensible heat net flux and the difference between air and surface skin temperatures. LAI data from the MODIS (Moderate
Resolution Imaging Spectroradiometer) product MCD15A3H.006 (Myneni et al., 2015) with a 500 m resolution were aggregated to a 0.25 degree scale using Google Earth Engine to be consistent with the GLDAS climatic drivers. Missing data were linearly interpolated, and a Savitzky–Golay filter (Savitzky and Golay, 1964) was applied to diminish high-frequency noise in the LAI time series. To estimate $V_{cmax,w}$, a PFT map from the GLDAS land cover map derived from MODIS was used (Fig. S2). The $V_{cmax,w}$ of each PFT was set as the static PFT-average from Walker et al. (2017) and corrected by temperature following
Medlyn et al. (2002). The maximum rooting depth was obtained from a global map synthesized from *in situ* observations (Fan et al., 2017). Soil texture from the Harmonized World Soils Database (FAO/IIASA/ISRIC/ISSCAS/JRC, 2012) was used to calculate soil drainage parameters based on empirical relations (Clapp and Hornberger, 1978).



## 2.5 Analyses

### 2.5.1 Observing system simulation experiment

To test the capability of the model-data fusion approach to correctly retrieve parameters under the presence of observational noise, we conducted an Observing System Simulation Experiment (OSSE) for 50 pixels. The 50 pixels were randomly distributed across the globe. The OSSE uses synthetic rather than real observations to test data-assimilation uncertainty, among other objectives (Arnold and Dey, 1986; Nearing et al., 2012; Errico et al., 2013). At each pixel, time series of VOD, surface soil moisture and ET were generated by using the model (Section 2.1) with prescribed parameters. To mimic the presence of

observational noise in real observational estimates, white noise was then added to the simulated values of VOD, surface soil moisture ET. The prescribed standard deviations of noise in VOD, surface soil moisture and ET, i.e., 0.05, 0.08, and 0.5 mm day$^{-1}$ respectively, were chosen to be within the mid 50% ranges retrieved using real data. The parameters retrieved using the model-data fusion approach were then compared with the prescribed values.

### 2.5.2 Comparison between derived traits and *in situ* measurements

Because hydraulic traits are often measured at a single plant or a segment scale that is much smaller than the ecosystem scale used in model-data fusion, and because of the relatively coarse spatial resolution of the remote sensing data used as constraints here, a one-to-one comparison between *in situ* data and model-data fusion derived values is likely to be dominated by representativeness error. Instead, we aggregated both *in situ* measurements and the traits derived here by PFTs to evaluate whether across-PFT patterns can be captured. Among the most ecologically important and widely measured traits are $g_1$ and

$\psi_{50,x}$, which indicate stomatal marginal water use efficiency and vulnerability to xylem cavitation, respectively. Synthesized datasets of $g_1$ from Lin et al. (2015) and $\psi_{50,x}$ from Kattge et al. (2011) based on *in situ* measurements covering a variety of species and climate types were used for comparison. In addition, Trugman et al. (2020) derived a map of tree $\psi_{50,x}$ across the continental United States at a 1 degree resolution, which integrated measurements in the Xylem Functional Trait Database and the US Forest Service Forest Inventory and Analysis (FIA) long-term permanent plot network. This map was used for a pixel-

wise comparison with the $\psi_{50,x}$ retrieved here in US areas dominated by forests. To perform this comparison, our model-data fusion derived traits were first aggregated from 0.25° to the 1° resolution of the estimates by Trugman et al. (2020).

### 2.5.3 Clustering analysis

To understand the global pattern of retrieved plant hydraulic traits, we constructed hydraulic functional types (HFTs) using the K-means clustering method (MacQueen, 1967). This method classifies each pixel to the nearest mean, i.e., the cluster center

in the five dimensional space spanned by the modeled hydraulic traits. To find the optimal number of clusters, we calculated the ratio between the variance within an across clusters traits across 3 to 20 clusters. The elbow method was used to derive the optimal number of clusters (Kodinariya and Makwana, 2013). That is, the optimal number of clusters was chosen based on the inflection point (elbow) of the curve relating the above ratio and the number of clusters. The global pattern of these





HFTs were examined. To provide insight into whether HFTs could be used as an alternative to PFTs, we evaluated how much
the accuracy of estimated VOD and ET would degrade if VOD and ET were modeled using hydraulic traits based on an HFT-
based clustering rather than a more typical PFT-based clustering. That is, we calculated the simulated VOD and ET by assigning
hydraulic traits as the center values for the HFT present at each pixel, rather than by using the average derived value across
each PFT as the PFT-wide value. Several factors differ between this calculation and the potential reduced error from using
HFTs in land surface models. For example, land surface models often use sub-grid scale tiling systems that are more complex
than the pixel-scale calculations performed here. The calculation here also did not account for uncertainties in determining the
optimal PFT-wide or HFT-wide values, or indeed, the mapping of PFTs or HFTs to begin with (Poulter et al., 2011; Hartley
et al., 2017). Nevertheless, this analysis provides first order insight into the capacity of HFT-based parametrization to improve
over a PFT-based approach.

## 3  Results

### 3.1  Parameter retrieval in the OSSE

Across the 50 pixels tested in the OSSE, the prescribed traits can be recovered using model-data fusion, with high correlations
between the assumed and retrieved values (Fig. 1). The hydraulic traits of $g_1$, $\psi_{50,x}$, and $g_{p,max}$ along with the soil parameters
($b_0$ in Eq. 4 and the boundary condition $b_c$) are accurately recovered ($r \geq 0.77$). The $C$ and the ratio between $\psi_{50,s}$ and $\psi_{50,x}$
showed larger discrepancies and greater uncertainty ranges due to the presence of (simulated) observational noise. For all
parameters, the residual errors are randomly distributed, rather than scaling the the true parameter value. Overall, the OSSE
supports the effectiveness of the model-data fusion approach.

### 3.2  Accuracy of modeled VOD, ET, and surface soil moisture

Over the entire study period of 2003-2011, the coefficient of determination ($R^2$) between estimated and observed VOD has
a median of 0.38 and a mid-50% range of (0.22,0.55) across the globe (Fig. 2a). The estimated VOD is highly correlated
with observations in northern and southwestern Australia, northeastern China, India, central Europe, Africa, and eastern South
America. The high VOD accuracy in these areas is likely partially a result of the large contribution of biomass to VOD due to
strong biomass seasonality in these areas (Liu et al., 2011; Momen et al., 2017). Notably, however, even in areas where VOD
has been shown to be less correlated with LAI, including central Australia, central Asia, South Africa, and the western US
(Momen et al., 2017), the estimated VOD accounting for the signature of leaf water potential is also able to capture observed
VOD. The model also accurately estimates observed ET with a median $R^2$ of 0.60 and a mid-50% range of (0.36,0.78) (Fig.
2b). Unlike in the majority of the world, the $R^2$ of ET is relatively lower in central Australia, southern South America, and
the southwestern US, where highly heterogeneous vegetation cover such as savannas and coexisting grass and shrubs within
a pixel could undermine model accuracy. The median and mid-50% range of surface soil moisture $R^2$ is 0.22 and (0.08,0.42),
respectively. Modeled surface soil moisture is less accurate in croplands (likely due to irrigation), as well as in boreal regions,



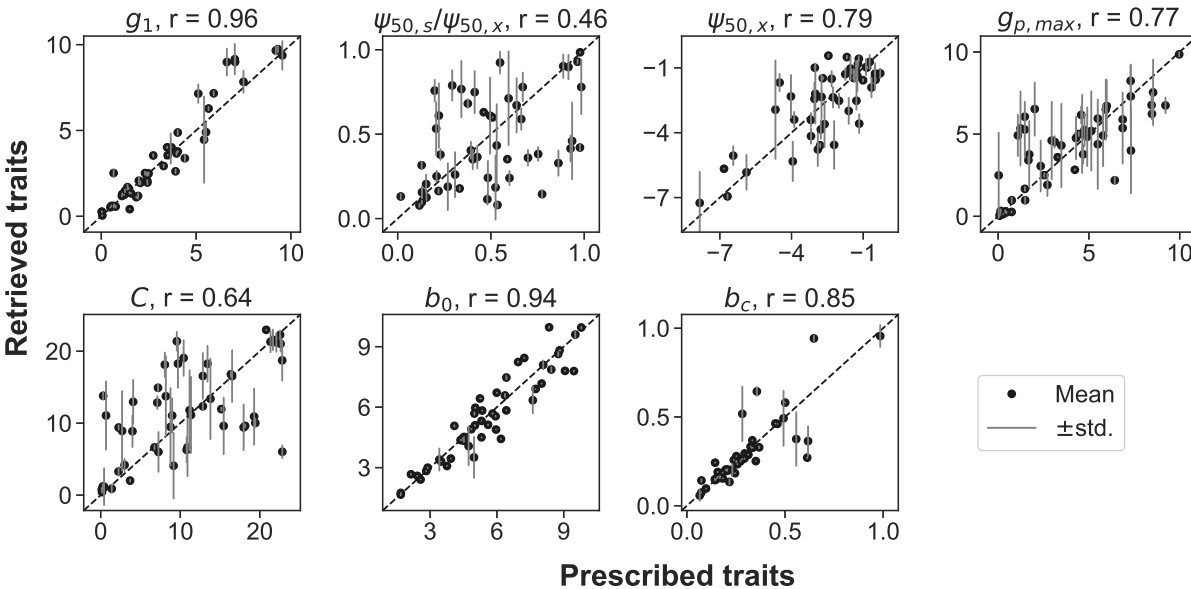

**Figure 1.** Comparison between the prescribed and retrieved plant hydraulic traits ($g_1$, $\psi_{50,s}/\psi_{50,x}$, $\psi_{50,x}$, $g_{p,max}$, $C$) and soil properties ($b_0$, $b_c$) in the observing system simulation experiment. The black dots and grey lines represent the mean and range of one standard deviation of retrieved posterior distributions. The diagonal dashed line is the 1:1 line. Correlation coefficient ($r$) between the prescribed and retrieved parameters are noted.

eastern China, Europe, and the mid-west and east of US. These regions largely overlap with those where the observed soil moisture from AMSR-E is weakly correlated with the reanalysis product of ERA-Interim that integrates ground observations (Parinussa et al., 2015), suggesting greater uncertainties of surface soil moisture from AMSR-E compared to other regions. The overall accuracy of estimated VOD, ET and surface soil moisture both within (Fig. S3) and outside (Fig. 2) the training period 2004-2005 suggest that the model and the derived traits effectively represent plant hydraulic dynamics.

**3.3   Global pattern of plant hydraulic traits**

The retrieved stomatal conductance slope parameter $g_1$, which is inversely proportional to marginal water use efficiency (Eq. 6), exhibits clear spatial patterns (Fig. 3a). High $g_1$ values arise in areas covered by grasses and savannas, such as the western US, the Sahel, central Asia, northern Mongolia, and inner Australia. This pattern is consistent with predictions from experimental data and optimality theory that herbaceous species – given the low cost of stem wood construction per unit water transport –

should have the largest $g_1$, i.e., be the least water-use efficient (Manzoni et al., 2011; Lin et al., 2015). In addition, croplands in India and eastern China also show high $g_1$, consistent with the high isohydricity of these regions (Konings and Gentine, 2017). Consistent with ground measurements that suggest $g_1$ increases with biome average temperature (Lin et al., 2015), the $g_1$ derived here is also (on average) lower in boreal ecosystems than in temperate and tropical ecosystems.

**Figure 2.** Assimilation accuracy ($R^2$) of (a) VOD, (b) ET, and (c) soil moisture during the entire study period 2003-2011. Insets show the probability distribution (pdf) of $R^2$ across the entire study area. Gray shaded area is not included in analysis.



Highly negative $\psi_{50,x}$ values are found in boreal evergreen needleleaf forests and in arid or seasonally dry biomes covered
by forests, shrubs or savannas, such as the western US, Central America, eastern south America, southeastern Africa, and
Australia (Fig. 3b). However, $\psi_{50,x}$ is more spatially scattered than $g_1$. This could partially arise from the greater coefficient
of variation across ensembles of $\psi_{50,x}$ (Fig. 4), suggesting $\psi_{50,x}$ is less tightly constrained compared to $g_1$ (consistent with
site-scale model-data fusion efforts in Liu et al. (2020) and the uncertainty estimates in the OSSE, Fig. 1). This additional
uncertainty might translate to more 'noise' in the ensemble medians for $\psi_{50,x}$ than that for $g_1$. Maps of other hydraulic traits
are shown in Fig. S4. The patterns of hydraulic traits exhibit greater variability beyond PFT distribution (Fig. S2) and only
limited correlation with soil and climate conditions (Fig. S5).

Among the plant hydraulic traits, we found strong coordination between the vulnerability of stomata and the xylem ($\psi_{50,s}$
and $\psi_{50,x}$) across space (Fig. S5), consistent with existing evidence from ground measurements (Anderegg et al., 2017). Other
hydraulic traits are only weakly correlated, including $g_{p,max}$ and $\psi_{50,x}$ (Fig. S5), which is consistent with the previous finding
suggesting the safety-efficiency trade-off of xylem traits is weak across > 400 species (Gleason et al., 2016).

Across PFTs, evergreen needleleaf forests have the lowest $g_1$, followed by deciduous broadleaf forests and shrublands (Fig.
5a). Grasslands and croplands have the highest $g_1$. This trend follows the across-PFT pattern found by (Lin et al., 2015). The
estimated across-PFT pattern of mean $\psi_{50,x}$ is also consistent with measurements included in the TRY database (Kattge et al.,
2011), i.e., lowest in grasslands and highest in evergreen needleleaf forests (Fig. 5b). However, across the globe, we found the
average standard deviation within PFTs is 3.6 and 2.3 times the standard deviation across PFTs for $g_1$ and $\psi_{50,x}$, respectively.
The large within-PFT variation is consistent with *in situ* observations (Anderegg, 2015), indicating PFTs are not informative
of plant hydraulic traits.

We further compared the retrieved $\psi_{50,x}$ for specific locations to an alternative estimate upscaled from forest inventory
(FIA) surveys (Fig. 6). Consistent with the FIA-based estimate, the retrieved $\psi_{50,x}$ are overall lower in pixels dominated by
evergreen needleleaf forests than in evergreen and deciduous broadleaf forests and mixed forests. However, across pixels, the
ecosystem-scale $\psi_{50,x}$ derived from remote sensing vary significantly more than the estimates from the Trugman et al. (2020)
dataset. Some fraction of this discrepancy might be due to intra-species variability in $\psi_{50,x}$, which is not accounted for in the
FIA-based estimate, and due to uncertainty in the kriging-based interpolation used for upscaling from the sparse FIA plots
to each $1°$ pixel. Nevertheless, this discrepancy highlights the scale-gap between traits measured for a single plant and that
derived for an ecosystem.

## 3.4 Hydraulic functional types (HFTs)

We built six HFTs (termed H1 to H6) using the K-means clustering method. The number of clusters (six) was chosen using the
elbow method based on the inflection point of the ratio of within- to across-clusters variance (Fig. S6). Across the six HFTs,
the across-cluster variance is 1.7 times as large as the within-cluster variance. The HFTs explain 57% of the total variance of
hydraulic traits across the globe. The cluster centers of the six HFTs are characterized by distinct combinations of hydraulic
traits (Fig. 7a). Specifically, H1 and H2 feature low $\psi_{50,s}$ and $\psi_{50,x}$, and are mainly distributed in boreal forest and arid or
seasonally dry biomes including the western US, Central America, southeastern Africa, central Asia, and Australia (Fig. 7b).



a

$g_1$ (kPa$^{1/2}$)

b

$\psi_{50,x}$ (MPa)

**Figure 3.** Global maps of (a) $g_1$ and (b) $\psi_{50,x}$ retrieved using model-data fusion. Posterior mean of each pixel is plotted.





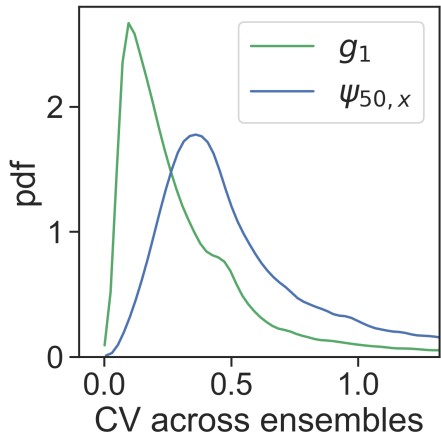

**Figure 4.** Empirical distribution across pixels of the coefficient of variation (CV) of $g_1$ and $\psi_{50,x}$ calculated across ensembles.

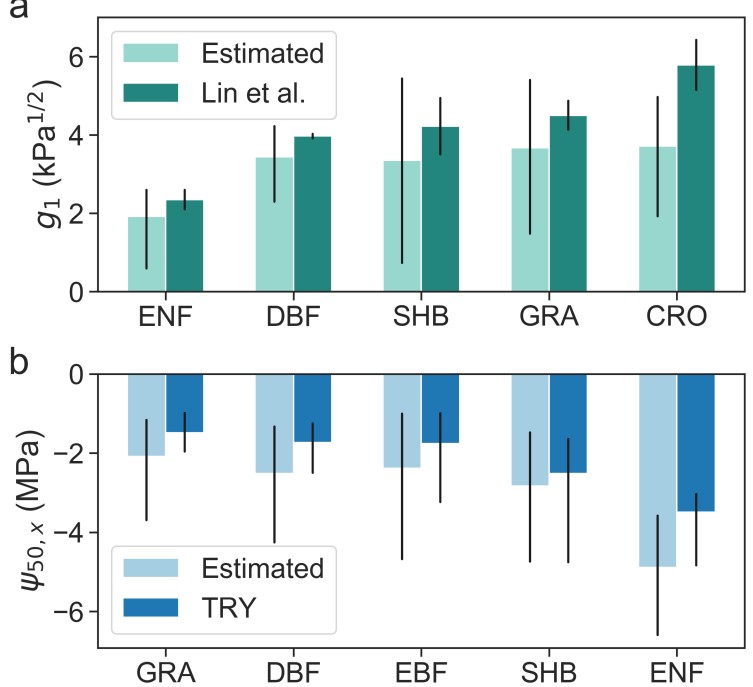

**Figure 5.** Retrieved (a) $g_1$ and (b) $\psi_{50,x}$ using model-data fusion (light colored bars) grouped by PFTs, in comparison with values derived from in situ measurements (dark colored bars) reported in Lin et al. (2015) and the TRY database (Kattge et al., 2011). Compared PFTs include evergreen needleleaf forest (ENF), deciduous broadleaf forest (DBF), evergreen broadleaf forest (EBF), shrubland (SHB), grassland (GRA), cropland (CRO). Bars represent medians of each PFT and black lines indicate the 25th-75th percentile ranges. The $g_1$ averaged across gymnosperm trees and angiosperm trees from Lin et al. (2015) were compared to retrieved $g_1$ in pixels dominated by ENF and DBF, respectively.





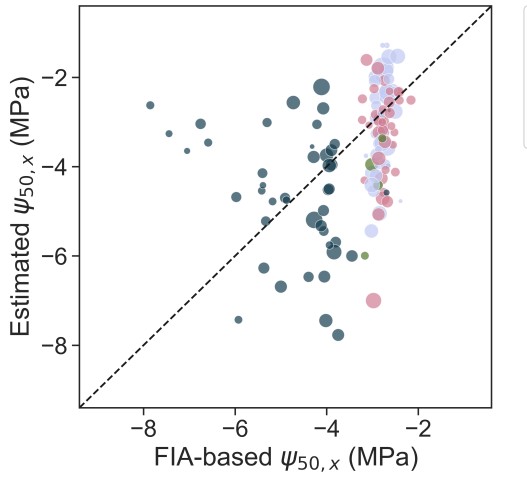

**Figure 6.** Aggregated $\psi_{50,x}$ from upscaling Forest Inventory and Analysis (FIA) plots based on Trugman et al. (2020) and $\psi_{50,x}$ retrieved here for corresponding pixels. The point size is scaled by number of plots used in aggregation for each pixel.

H3 and H4 are characterized by low and high vegetation capacitance ($C$) respectively, though both have low $g_{p,max}$. H3 is mainly but not exclusively distributed in grasslands and savannas in the central US, the Nordeste region in Brazil, eastern South Africa, and eastern Australia, as well as in the Miombo woodlands. H4 is distributed in shrublands in the southwestern US, Argentina, southern Africa, northwestern India, and northeastern Australia. H5, often found in tropical and sub-tropical regions, is characterized by large $g_{p,max}$ and capacitance. H6 is characterized primarily by high $g_1$, which includes croplands in Indian, southeastern Asia, and central and eastern China. Note that the pattern of HFTs (Fig. 7b) is substantially distinct from the distribution of PFTs (Fig. S2), illustrating the limitations of parameterizing plant hydraulics based on PFTs.

Using averaged traits per PFT instead of pixel-specific traits to calculate VOD and ET led to a median increase in normalized root-mean-square-error (nRMSE, with the long-term average used for normalization) of 0.82 and 0.58, respectively. This degradation of accuracy is unsurprising given the high spatial variability of hydraulic traits and the fact that PFTs are not categorized specifically to distinguish plant hydraulic functions. However, using the hydraulic traits averaged per HFT instead improves prediction accuracy over the PFT-based predictions. When compared to using pixel-specific values, using average traits based on HFTs increases the nRMSE by 0.65 and 0.42 for VOD and ET, respectively. In each case, this is less than the degradation when PFT-based averages are used. Indeed, when PFT-based instead of HFT-based model estimates are compared, the nRMSE of ET increases by more than 0.1 in 58% of the analyzed area (Fig. 8a). ET is mainly improved in arid or seasonally dry biomes, including the western US, southern South America, southern and eastern Africa, central Asia, and Australia. In addition, the normalized RMSE of VOD is also improved by more than 0.5 in 37% of the analyzed area using HFTs rather than PFTs (Fig. 8b). Areas exhibiting reduced error are mainly located in the southwestern US, Central America, eastern South America, Mediterranean, Africa and Australia, where variation of leaf water potential has a strong signature on VOD (Momen



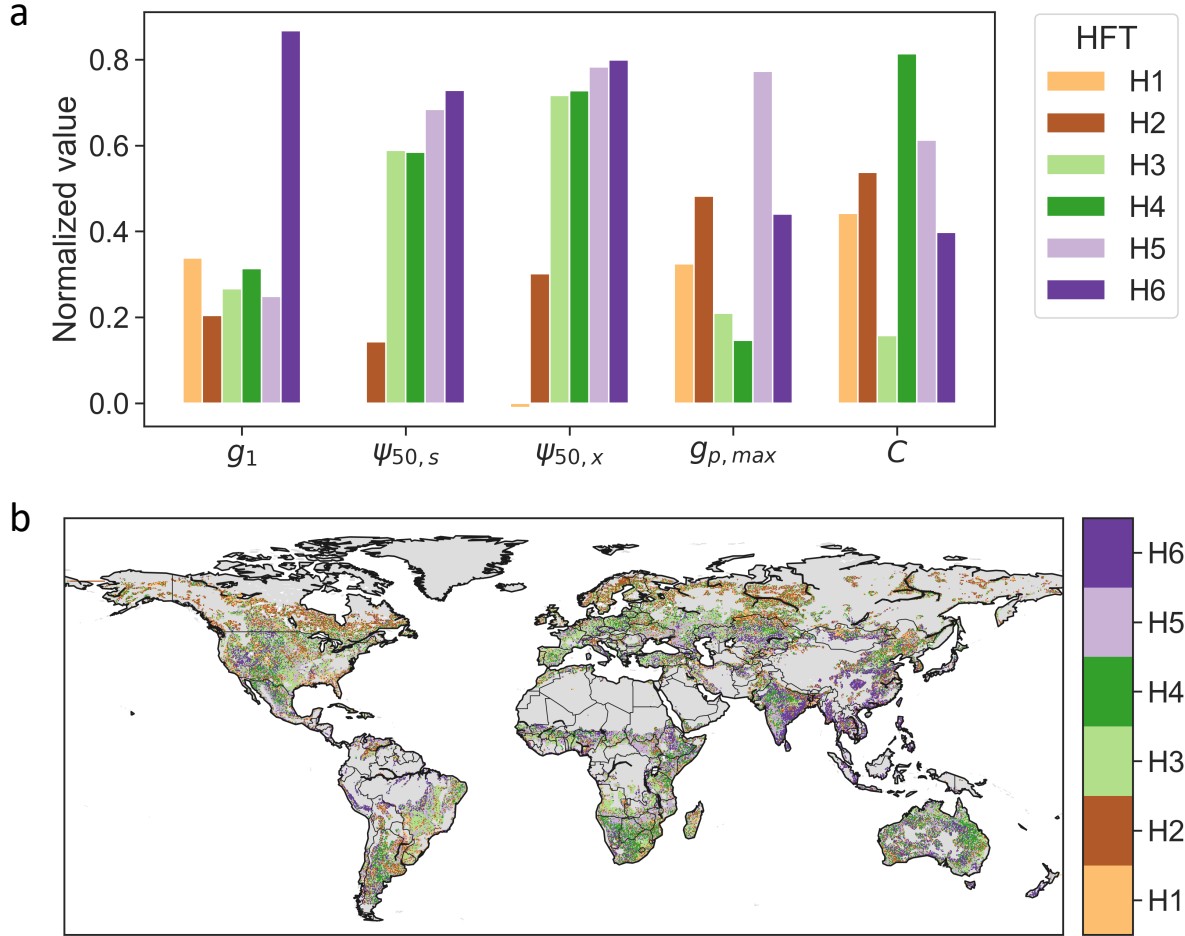

**Figure 7.** (a) Plant hydraulic traits of the centers of six hydraulic functional types and (b) their spatial pattern. Each trait of cluster centers is normalized using $(V - V_5)/(V_{95} - V_5)$, where $V$ is the trait magnitude, and $V_5$ and $V_{95}$ are the 5th and 95th percentiles of the corresponding trait across the study area.





et al., 2017). These findings suggest the importance of appropriate parameterization of hydraulic traits on capturing leaf water potential and ET variations at an ecosystem scale.

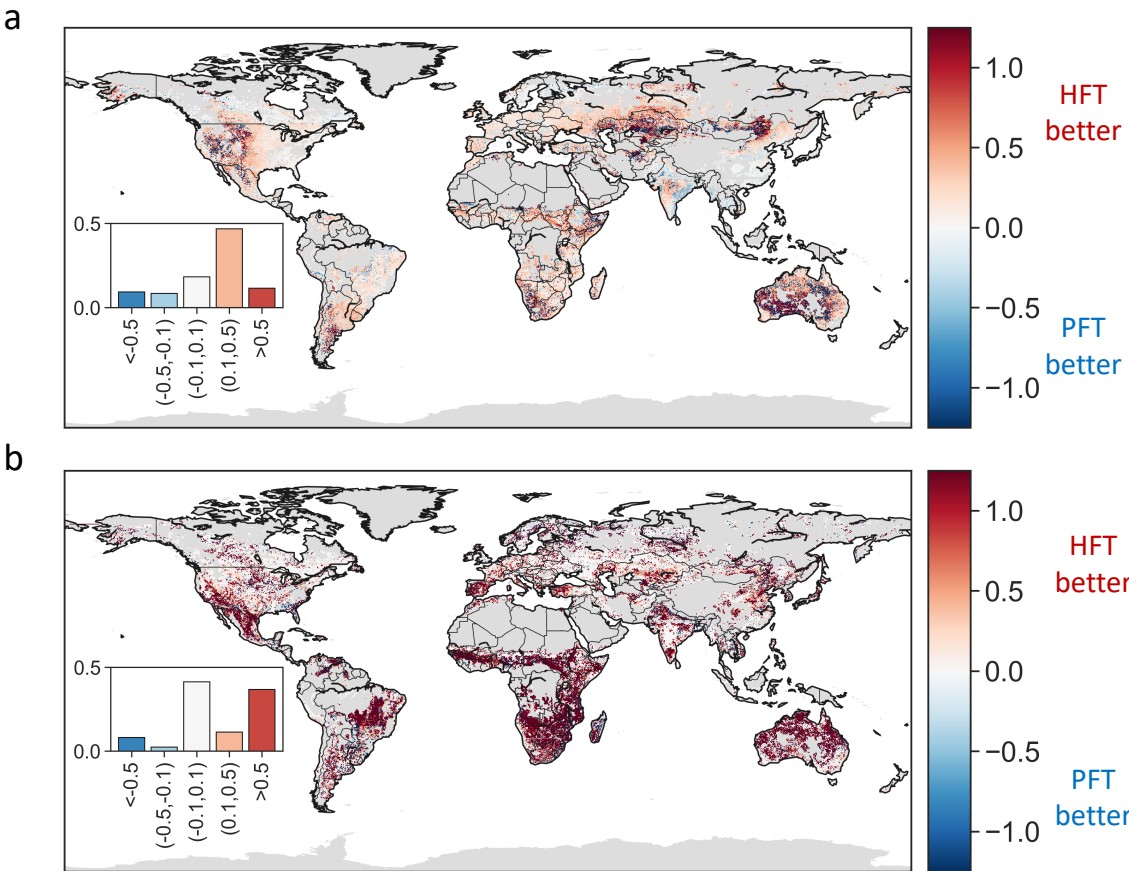

**Figure 8.** Normalized root mean square error (nRMSE) of estimated (a) ET and (b) VOD using traits averaged by plant PFTs minus that using traits averaged by HFTs. The insets show the areal frequency of the nRMSE difference.

## 4  Discussion

This study derived ecosystem-scale plant hydraulic traits across the globe using a model-data fusion approach. The retrieved traits enable the hydraulic model to capture plant hydraulic dynamics as reflected by ET and VOD from microwave remote sensing. While the traits derived here are consistent with across-PFT patterns based on *in situ* measurements, they also exhibit large within-PFT variations. Using specific 'hydraulic functional types' rather than PFTs as a clustering mechanism for deriving average traits improves the accuracy of estimated ET and VOD even as the number of functional types (each of which has

unknown optimal values in an applied modeling setting) is reduced. We note that the exact values of hydraulic traits depend on

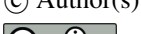



the specific data and model representation used here and therefore are subject to model and data uncertainties. However, the derived spatial pattern of traits and other main findings highlight opportunities and challenges for further investigation of plant hydraulics at a global scale.

### 4.1 Contribution of VOD to informing plant hydraulic behavior

The fact that VOD varies with plant water content allows investigation of plant physiological dynamics at a large scale. Although VOD has often been used as a proxy of aboveground biomass (e.g. Liu et al., 2015; Tian et al., 2017; Brandt et al., 2018; Teubner et al., 2019), it is in fact determined by both biomass and plant water status (Konings et al., 2019). VOD variations within a day (Konings and Gentine, 2017; Li et al., 2017; Anderegg et al., 2018) and during soil dry-downs (Feldman et al., 2018; Zhang et al., 2019; Feldman et al., 2020) highlight the sensitivity of VOD to relative water content. At seasonal and inter-

annual scales, VOD has also been found to be modulated by leaf water potential or relative water content, thus deviating from biomass signals (Momen et al., 2017; Tian et al., 2018; Tong et al., 2019). Here, after parsing out the impact of biomass through LAI, VOD provides information about leaf water potential variation and therefore contributes to constraining the underlying hydraulic traits. Kumar et al. (2020) previously assimilated VOD into a land surface model as a constraint on biomass, which led to improvements in modelled ET. Our findings suggest that when assimilated into models with an explicit representation of

plant hydraulics, VOD can act to constrain both water and carbon dynamics and their respective climatic responses. Although not explored in detail in this study, note also that, by determining optimal values for $a$, $b$, and $c$, (the parameters relating VOD to $\psi_l$ in Eq. 11), the model-data fusion system introduced here also allows determination of $\psi_l$ from VOD, which may be of interest for a variety of studies of plant responses to drought.

Our previous study (Liu et al., 2020) at a stand-scale has shown that stomatal traits are well-constrained using ET alone,
whereas xylem traits including $\psi_{50,x}$ remain largely under-constrained, in part due to lack of information on leaf water potential. Incorporating VOD among the constraints here contributes to separation of xylem and stomatal behavior. As a result, the model-data fusion approach here is, to our knowledge, the first to be able to retrieve both stomatal and xylem traits across the globe. Nevertheless, $\psi_{50,x}$ is still less well-resolved across ensembles compared to other traits (Fig. 4). This could result from trade-offs among hydraulic traits and the lack of constraints on the scaling from leaf water potential to VOD, which varies

across space. More prior information about these two factors will likely contribute to improved retrieval of plant hydraulic traits. Additionally, the use of solar-induced fluorescence or other constraints on photosynthesis may allow for independent information about stomatal closure that could be used to improve the accuracy and certainty of the retrieved hydraulic traits. However, care should be taken that the uncertainty introduced by coupling to a photosynthesis model does not outweigh the added advantage of this additional constraint.

### 4.2 Bridging the spatial scale gap of hydraulic traits

Plant hydraulic traits vary among segments from root to shoot even for a single tree, causing the hydraulic sensitivity at a whole-tree scale to be distinct from that measured at a segment scale (Johnson et al., 2016). Likewise, species diversity, canopy structure, and demographic composition can cause large variability of hydraulic traits. As a result, a community-weighted





average of a trait may not well represent the integrated hydraulic behavior at an ecosystem scale, as evidenced, for example, by
the significant effect of plant hydraulic diversity on evapotranspiration responses to drought (Anderegg et al., 2018). Here, we
also found substantial discrepancy between community-weighted $\psi_{50,x}$ and the ecosystem-scaled value derived representing
the property of the entire pixel, even in the most extensively surveyed pixels available (biggest dots in Fig. 6). This highlights
the challenge of scaling up ground measurements of plant hydraulic traits to a scale relevant to land surface modeling from
the bottom-up. The model-data fusion used here provides an approach to help address this challenge. However, further study is
needed to explore how stand and ecosystem characteristics shape the ecosystem-scale hydraulic traits, as well as the effective
relationship between leaf water potential and remote-sensing scale water content.

### 4.3 Implications for land surface models

Because they are able to predict ET and VOD better than PFTs (Fig. 8), the HFTs point to the potential for a better parameteri-
zation scheme of plant hydraulics in land surface models. Because HFTs require fewer clusters than PFTs do to model ET with
the same or better accuracy, parameterizing plant hydraulics by HFTs in land surface models may contribute to higher model
accuracy. However, because the magnitude of state variables may differ between models even as their temporal dynamics don't
(Koster et al., 2009), including between a give land surface model and the model used here, using the exact values derived here
may cause errors. Instead, the map of HFTs and their relative magnitude of traits can be used as a baseline for model-specific
calibration. Moreover, moving beyond fixed values for each HFT, hydraulic traits within each type may be further related to to
landscape features such as climate, topography, canopy height and stand age using the environmental filtering approach (Butler
et al., 2017). As demonstrated for photosynthetic traits (Verheijen et al., 2013; Smith et al., 2019), such relationships allow
practical flexibility to account for trait variations across space, thus improving the performance of large-scale models. They
may also allow improved compatibility with sub-grid tiling schemes used by land surface schemes. As land surface models
that explicitly represent plant hydraulics are becoming more common, our results demonstrate the possibility of alternative,
computationally efficient approaches to parameterizing plant hydraulic behavior, which will contribute to improved prediction
of natural resources and climate feedbacks.



*Code and data availability.* All the datasets used in this study are publicly available from the referenced sources, except that microwave-based ALEXI ET was obtained upon request to Thomas R. Holmes and Christopher R. Hain on January 28, 2020. The source code of the used plant hydraulic model and the model-data fusion algorithm is available at https://github.com/YanlanLiu/VOD_hydraulics. The maps

of retrieved ensemble mean and standard deviation of plant hydraulic traits are publicly available on Figshare https://doi.org/10.6084/m9. figshare.13350713.v1

*Author contributions.* YL, NH and AGK conceived the study. YL and NH set up the model. YL prepared data and conducted the analyses, with all the authors providing input. YL led the manuscript writing. All the authors contributed to editing the manuscript.

*Competing interests.* The authors declare no competing interests.

*Acknowledgements.* We thank Thomas R. Holmes and Christopher R. Hain for providing the ALEXI ET data. The authors were supported by NASA Terrestrial Ecology award 80NSSC18K0715 through the New Investigator Program. AGK was also supported by NOAA grant NA17OAR4310127. NH is supported by NASA through FINESST (Future Investigators in NASA Earth and Space Science and Technology) grant 19-EARTH20-0078.





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
