# Peer review of "Global ecosystem-scale plant hydraulic traits retrieved using model-data fusion"

_Hydrology and Earth System Sciences, 2020_

## Referee Comment (RC1) · Anonymous Referee #1 · 12 Jan 2021

The study introduces the concept of hydraulic functional types, which represent distinct combinations of plant hydraulic traits and which may contribute to an improved parameterization of vegetation in land surface models. It is further evaluated if hydraulic functional types can present an alternative to commonly used plant functional types.

The study thus presents indeed new insight into the spatial distribution of certain trait combinations that might be relevant for further applications in global vegetation analysis. The manuscript fits well into the scope of the journal and might attract readers interested in vegetation modeling.

The manuscript is well structured and the methods are described appropriately. I only have minor comments.

Minor comments:

- Define which correlation coefficient was used - assuming it was the Pearson correlation coefficient?

- Information about the PFT-names should be included in the main part when PFTs are first mentioned.

- Add a conclusion section referring to the main results.

———————————————

---

## Referee Comment (RC2) · Anonymous Referee #2 · 19 Feb 2021

This study presents an intergrated approach between vegetation modeling and Earth observations to derive the hydraulic trait, a crucial parameter to better understand the impact of droughts on vegetation. This new approach is likely to improve the parameterization of vegetation in land surface models. The manuscript is well organized and written, the approach is well described, the results are convincing. It also fits the scope of the journal. I just have a few minor comments:

1) In the abstract and other parts of the manuscript, you write: "VOD is proportional to vegetation watercontent and therefore closely related to leaf water potential". You should specify that you choose a high-frequency VOD, X-band instead of L-band for instance, as the high frequencies are more sensitive to the leafy component of the vegetation and the lower frequencies to the woody component (see Frappart et al.,

Remote Sensing, 2020 for a recent review on VOD).

2) Why did not you include the equatorial and boreal forests in this study? It would have been interesting to seee the results in Amazonia to compare them with the gradient observed by Jones et al., Environmental Research Letters, 2014, who identify adaptation to both water and light availability from west to east.

3) Why did you choose LPRM VOD and soil moisture products instead of LPDR? In a recently published study (Li et al., Remote Sensing of Environment, 2021) one of you is co-author, LPRM X-VOD was found to be more correlated to vegetation indices (NDVI, EVI, ...) than the other X-VOD products inluding LPRM.

---

## Author Comment (AC1) · 1 Mar 2021

Reviewer 1

The study introduces the concept of hydraulic functional types, which represent distinct combinations of plant hydraulic traits and which may contribute to an improved parameterization of vegetation in land surface models. It is further evaluated if hydraulic functional types can present an alternative to commonly used plant functional types. The study thus presents indeed new insight into the spatial distribution of certain trait combinations that might be relevant for further applications in global vegetation analysis. The manuscript fits well into the scope of the journal and might attract readers interested in vegetation modeling. The manuscript is well structured and the methods are described appropriately. I only have minor comments.

**Authors:** We thank the reviewer for providing positive and constructive comments. The response to each individual comment, along with associated edits to the manuscript text, are listed below in black.

- Define which correlation coefficient was used - assuming it was the Pearson correlation coefficient?

**Authors:** We now clarify that the Pearson correlation coefficient was used in both the results section (line 294) and the caption of Fig. 1.

- Information about the PFT-names should be included in the main part when PFTs are first mentioned.

**Authors:** Thanks for the suggestion. In the introduction (lines 40-42), where PFTs are first mentioned, we added "At large scales, plant traits are often parameterized based on plant functional types (PFTs), such as evergreen needleleaf forests, evergreen broadleaf forests, deciduous broadleaf forests, mixed forests, shrublands, grasslands and croplands".

- Add a conclusion section referring to the main result

**Authors:** We added a conclusion section at the end of the manuscript to summarize the main result, as follows:

"This study derived ecosystem-scale plant hydraulic traits across the globe using a model-data fusion approach. The retrieved traits enable our hydraulic model to capture the dynamics of leaf water potential and ET, based on comparison to remote sensing observations. While the traits derived here are consistent with across-PFT patterns based on in situ measurements, they also exhibit large within-PFT variations (as expected). There is some discrepancy between our derived $\psi_{50,x}$ and values derived from interpolating between forest inventory plots, though it is unclear if this discrepancy is caused by errors in the model-data fusion retrievals, errors in the upscaled inventory data due to intra-specific variability and spatial interpolation imperfections, or both. Uncertainty is also induced by whether or not our retrievals represent the same effective values as a community-weighted average (see Section 4.2). Nevertheless, reasonable correspondence between the across-PFT variations in our derived traits compared to in situ measurements add confidence to the dataset introduced here.

As an alternative to PFTs, we constructed "hydraulic functional types" based on clustering of the derived hydraulic traits. Using the hydraulic functional types, rather than PFTs, to drive averaged traits by functional types improves the accuracy of estimated ET and VOD, even as the number of functional types is reduced relative to a PFT-based representation. This suggests that hydraulic functional types may form a computationally efficient yet promising approach for representing the diversity of plant hydraulic behavior in large-scale land surface models. We note that the exact values of the derived hydraulic traits depend on the specific data and model representation used here and therefore are subject to model and data uncertainties. However, our findings highlight opportunities and challenges for further investigation of plant hydraulics at a global scale."

---

## Author Comment (AC2) · 1 Mar 2021

Reviewer 2

This study presents an intergrated approach between vegetation modeling and Earth observations to derive the hydraulic trait, a crucial parameter to better understand the impact of droughts on vegetation. This new approach is likely to improve the parameterization of vegetation in land surface models. The manuscript is well organized and written, the approach is well described, the results are convincing. It also fits the scope of the journal. I just have a few minor comments:

**Authors:** We thank the reviewer for providing positive and constructive comments. The response to each individual comment, along with associated edits to the manuscript text, are listed below in black.

1) In the abstract and other parts of the manuscript, you write: "VOD is proportional to vegetation water content and therefore closely related to leaf water potential". You should specify that you choose a high-frequency VOD, X-band instead of L-band for instance, as the high frequencies are more sensitive to the leafy component of the vegetation and the lower frequencies to the woody component (see Frappart et al., C1 HESSD Interactive comment Printer-friendly version Discussion paper Remote Sensing, 2020 for a recent review on VOD).

**Authors:** In the methods (lines 141-142), we noted that X-band VOD was used. It is also specified that "This dataset is based on observations at X-band frequency (10.7 GHz), which is primarily sensitive to water content of the upper canopy layers (Frappart et al., 2020)." We have now added the following sentence for clarity (lines 142-144): "Here, we used an X-band record rather than lower microwave frequencies to reduce errors associated with potential sensitivities of these lower frequencies to xylem water potential, which might deviate from leaf water potential".

2) Why did not you include the equatorial and boreal forests in this study? It would have been interesting to see the results in Amazonia to compare them with the gradient observed by Jones et al., Environmental Research Letters, 2014, who identify adaptation to both water and light availability from west to east.

**Authors:** While we agree that the described comparison would have been interesting, parts of equatorial and boreal forests were removed if the long-term average VOD exceeds 0.8. We filtered such pixels to remove the effect of high retrieval uncertainty in these regions - at high VOD, there is little signal expected from the soil, but some soil signal is still assumed in the retrieval algorithm. This can have the effect of adding noise to the VOD. This filtering criterion is explained in the Methods section (lines 221-225): "Pixels where VOD is below 0.15 or above 0.8 were also excluded to remove sparsely vegetated pixels and extremely dense vegetation areas, respectively. The most densely vegetated areas were removed because low microwave transmissivity significantly reduces the accuracy of VOD and soil moisture retrievals there (Kumar et al, 2019), and low VOD pixels were removed to reduce inaccuracies due to ground volume scattering and low vegetation density."

3) Why did you choose LPRM VOD and soil moisture products instead of LPDR? In a recently published study (Li et al., Remote Sensing of Environment, 2021) one of you is co-author, LPRM

X-VOD was found to be more correlated to vegetation indices (NDVI, EVI, ...) than the other X-VOD products inluding LPRM.

**Authors:** Our application here is primarily interested in those components of the VOD timeseries that vary as a result of changes in water stress and associated leaf water potential changes. Thus, a greater correlation between VOD and optically-based vegetation indices representing canopy greenness and structure (e.g., canopy height) are not necessarily indicative that that particular dataset is a better fit for this particular application. Our choice to use LPRM instead of LPDR here was based on the fact that, as written in its documentation files, the LPDR product pre-applies a 30-day moving window average to the dataset to smooth out variability. This is much coarser than the 5-day moving window we applied to LPRM data and has the possible effect of reducing meaningful variations due to water stress.

However, we agree with the reviewer that there may be other differences between the algorithms that can affect the hydraulic traits derived here - for example, LPDR's use of a dynamic single-scattering albedo vs. LPRM's constant values. We had previously noted uncertainty associated with the choice of VOD dataset (lines 445-446): "the exact values of hydraulic traits depend on the specific data and model representation used here and therefore are subject to model and data uncertainties" (lines 386-387). To emphasize this point more, we have now added more text in the Discussion Section 4.1 to encourage further investigation on the impacts of different VOD products (lines 397-399): "However, additional research is needed to understand the effect of the choice of retrieval algorithm and specific VOD product (Li et al, 2021) on any inferred VOD-$\psi_l$ relationships. For this reason, any such efforts would also benefit from explicit uncertainty quantification."